# ACE Inhibitory Activity and Molecular Docking of Gac Seed Protein Hydrolysate Purified by HILIC and RP-HPLC

**DOI:** 10.3390/molecules25204635

**Published:** 2020-10-12

**Authors:** Samuchaya Ngamsuk, Tzou-Chi Huang, Jue-Liang Hsu

**Affiliations:** 1Department of Tropical Agriculture and International Cooperation, National Pingtung University of Science and Technology, 1 Shuefu Road, Neipu, Pingtung 91201, Taiwan; 2Department of Biological Science and Technology, National Pingtung University of Science and Technology, 1 Shuefu Road, Neipu, Pingtung 91201, Taiwan; tchuang@mail.npust.edu.tw; 3International Master’s Degree Program in Food Science, National Pingtung University of Science and Technology, 1 Shuefu Road, Neipu, Pingtung 91201, Taiwan; 4Research Center for Animal Biologics, National Pingtung University of Science and Technology, Pingtung 91201, Taiwan; 5Research Center for Tropic Agriculture, National Pingtung University of Science and Technology, Pingtung 91201, Taiwan

**Keywords:** gac seed protein, simulated gastrointestinal digestion, ACE inhibitory activity, LC-MS/MS, molecular docking

## Abstract

Gac (*Momordica cochinchinensis* Spreng.) seed proteins (GSPs) hydrolysate was investigated for angiotensin I-converting enzyme (ACE) inhibitory activities. GSPs were hydrolyzed under simulated gastrointestinal digestion using a combination of enzymes, including pepsin, trypsin, and chymotrypsin. The screening of ACE inhibitory peptides from GSPs hydrolysate was performed using two sequential bioassay-guided fractionations, namely hydrophilic interaction liquid chromatography (HILIC) and reversed-phase high-performance liquid chromatography (RP-HPLC). Then, the peptides in the fraction with the highest ACE inhibitory activity were identified by LC-MS/MS. The flow-through (FT) fraction showed the most potent ACE inhibitory activity when HILIC fractionation was performed. This fraction was further separated using RP-HPLC, and the result indicated that fraction 8 (RP-F8) showed the highest ACE inhibitory activity. In the HILIC-FT/RP-F8 fraction, 14 peptides were identified using LC-MS/MS analysis coupled with de novo sequencing. These amino acid chains had not been recorded previously and their ACE inhibitory activities were analyzed in silico using the BIOPEP database. One fragment with the amino acid sequence of ALVY showed a significant ACE inhibitory activity (7.03 ± 0.09 µM). The Lineweaver-Burk plot indicated that ALVY is a competitive inhibitor. The inhibition mechanism of ALVY against ACE was further rationalized through the molecular docking simulation, which revealed that the ACE inhibitory activities of ALVY is due to interaction with the S1 (Ala354, Tyr523) and the S2 (His353, His513) pockets of ACE. Bibliographic survey allowed the identification of similarities between peptides reported as in gac fruit and other proteins. These results suggest that gac seed proteins hydrolysate can be used as a potential nutraceutical with inhibitory activity against ACE.

## 1. Introduction

*Momordica cochinchinensis* Spreng. seeds have long been considered waste products in gac industry in Southeast Asian countries, such as Thailand, Laos, Myanmar, Cambodia, Vietnam, Malaysia, India, and Taiwan, where the fruit is widely grown [1]. However, in traditional Chinese medicine, gac seeds have been used as a treatment of certain diseases, such as diabetes, eye disorders, fluxes, liver spleen disorders, hemorrhoids, wounds, bruises, boils, sores, scrofula, tinea, swelling, and pus [2,3]. To explore more health benefits of gac seeds, this study aimed to investigate the antihypertensive peptides from enzymatic hydrolysate of gac seeds. Bioactive peptides can prevent oxidation and microbial degradation in foods and can be used for the treatment of various medical conditions, thus increasing the quality of life [4]. Bioactive peptides are liberated during proteolytic digestion of proteins and also during food processing (cooking, fermentation, and ripening) [5]. The enzymatic hydrolysis produces bioactive peptides more efficiently than microbial fermentation due to the short reaction time, ease of scalability, and predictability [6]. Moreover, the bioactivity of peptides has been suggested to depend mainly on the amino acid composition, sequence, structure, and other factions, such as hydrophobicity, charge, or even the binding properties of peptides [7,8].

Angiotensin I-converting enzyme (ACE) performs a critical role in blood pressure control within the renin-angiotensin system, which is the primary physiological pathway described for the control and management of blood pressure [9]. It catalyzes the cleavage of the C-terminal dipeptide from inactive angiotensin I to the active angiotensin II and also inhibits the activity of the vasodilator bradykinin [10,11]. Natural bioactive peptides from protein-rich foods provided an interesting explanation for ACE inhibition and have been explored extensively as a replacement of chemical drugs, such as captopril, enalapril, and lisinopril [12,13]. To date, there is no information on the potential of bioactive peptides from gac seed proteins (GSPs) as ACE inhibitors.

Solid-phase extraction (SPE) is a technique used mostly for sample pretreatment and enrichment [14]. SPE methodologies were developed for waste treatment and environmental monitoring. The development of new methodologies based on SPE made this technique more versatile, allowing pretreatment of any kind of sample in a wide concentration range. SPE is regarded as a separation method with advantages over other methods allowing a variety of applications along with speed, reproducibility, and efficiency [15].

Hydrophilic interaction liquid chromatography (HILIC) is an alternative separation tool for separating polar compounds [16]. HILIC has recently emerged as a popular chromatographic mode for the separation of hydrophilic analytes. HILIC operates on the basis of hydrophilic interactions between the analytes and the hydrophilic stationary phase with either highly polar or hydrophilic compounds interacting most strongly [17].

The BIOPEP-UWM database of bioactive peptides (formerly BIOPEP) has the potential for application of computational tools in peptide science based on a database [18,19,20,21,22,23]. It has recently become a popular tool in the research on bioactive peptides, especially on those derived from foods and being constituents of diets that prevent the development of chronic diseases [19].

The aim of this study was to identify bioactive peptides with potential ACE inhibitory activity from enzymatic hydrolysates of gac seed proteins. For this purpose, peptides in gac seed proteins hydrolysates were purified by HILIC and RP-HPLC and fractions were characterized by (I) the evaluation of its ACE inhibitory activity; (II) the identification of peptides (LC-MS/MS analysis followed by automatic de novo sequencing using PEAKS); (III) the prediction of peptides’ ACE inhibitory activities using BIOPEP in silico analysis; (IV) the identity and activity confirmations of the active peptides using synthetic peptides; and (V) the simulation of binding interaction between the identified peptides with ACE pockets.

## 2. Results and Discussion

### 2.1. IC_50_ Determination of GSPs Hydrolysate

GSPs were hydrolyzed using the combination of three proteolytic enzymes, including pepsin, trypsin, and chymotrypsin. The hydrolysate was passed through an ultrafiltration membrane with a molecular weight cutoff at 3 kDa (Millipore, MA, USA) followed by C_18_ SPE desalting. The IC_50_ of hydrolysate was determined using different concentrations to give a value of 70.0 ± 4 µg/mL, as shown in Figure 1. Based on the concentration of hydrolysate samples, similar results reported that the ACE inhibitory activity of neem (*Azadirachta indica)* seeds peptides was evidently increased with an increase in concentration of the peptide hydrolysate [24]. Similarly, studies reported that a triple-juice combination (pear, hemp seeds, and pumpkin seeds) exhibited higher ACE inhibition activity (IC_50_ of 82.54 µg/mL) than single (IC_50_ of 731.15 µg/mL) or double-juice combinations (IC_50_ of 245.72 and 117.67 µg/mL) [25]. A lower IC_50_ value corresponds to stronger inhibitory activity [26]. Many studies indicate that the antihypertensive activities of hydrolysates generated by the combination of two or more enzymes are higher than that hydrolyzed by solely an enzyme and the variation in their antihypertensive activity could be attributed to the differences in the composition and hydrophobicity of the protein primary structure [27,28]. Moreover, hydrophobic residues of amino acids (leucine, valine, alanine, tryptophan, tyrosine, proline, and phenylalanine) bind at the ACE’s catalytic site, acting as competitive inhibitors [29]. The results of the present study were in line with those reported in the literature and the ACE inhibitory activity of peptides at this concentration was high enough to be used in further experiments.

### 2.2. Bioassay-Guided Fractionation of GSPs Hydrolysate Using HILIC Purification

Hydrophilic interaction liquid chromatography (HILIC) emerged as an appealing chromatographic mode for the separation of polar compounds like small peptides [30]. HILIC uses hydrophilic stationary phases like those traditionally employed in the normal phase (NP) and also new specific ones [16]. The bioassay-guided fractionation of small peptides derived from GSPs hydrolysate was performed on HILIC. A total of eight different fractions were obtained and the ACE inhibitory activity of each fraction (at the same concentration) was examined using the in vitro ACE inhibitory assay. The result is shown in Figure 2. The highest ACE inhibitory activity among these eight fractions (each at 1 mg/mL) was exhibited by the flow-through (FT) fraction (78.20 ± 0.30%). The FT fraction of HILIC separation typically contained less polar peptides compared to those in other fractions, which implied hydrophobic peptides may exist in this fraction. In the case of antihypertensive activity, the results obtained in this study are similar to those reported in the literature [31]. Peptides derived from multiple enzymes showed high ACE inhibitory activities due to the possession of hydrophobic amino acids (leucine, valine, alanine, tryptophan, tyrosine, proline, and phenylalanine) and aromatic amino acids, which are typical features of ACE inhibitor peptides [28,29,32,33]. Therefore, it was expected that potential ACE inhibitory peptides were distributed in the HILIC flow-through fraction, which implied that further purification was necessary.

### 2.3. Further Bioassay-Guided Fraction of the HILIC FT Fraction Using RP-HPLC and ACE-Inhibitory Assay

To isolate the peptide showing the highest capability to inhibit ACE and therefore with the highest potential to prevent high blood pressure, the HILIC FT fraction of GSPs hydrolysate was further fractionated by RP-HPLC. After the optimization of the chromatographic method, 12 fractions were collected. Figure 3A depicts the chromatogram obtained for HILIC FT fraction and the fractions collected were labeled as F1 up to F12 and their respective percentages of ACE inhibitory activity were 1.32 ± 0.17, ~0.00, 0.76 ± 0.05, ~0.00, 21.79 ± 0.02, 12.91 ± 1.47, 21.10 ± 0.37, 25.63 ± 0.57, 6.84 ± 0.24, 0.86 ± 0.04, 17.79 ± 0.33, and 3.42 ± 0.32%. The highest inhibitory activity was expressed by fraction F8 as indicated in Figure 3B. Therefore, the fraction (HILIC-FT/RP-F8) was selected for further identification of potential ACE inhibitory peptides. 

### 2.4. Identification of Peptides in the HILIC-FT/RP-F8 Fraction Using LC-MS/MS Analysis Coupled with de Novo Sequencing

Since the HILIC-FT/RP-F8 fraction showed the highest ACE inhibitory activity, it may contain potent ACE inhibitory peptides. However, given that the genome of gac fruit (*Momordica cochinchinensis* Spreng.) has not been decoded completely, the protein database was not sufficient for comprehensive database-assisted peptide identification. Therefore, the peptide identification was performed using LC-MS/MS analysis and de novo sequencing on PEAK Studio 8.0. The LC-MS/MS chromatogram of the HILIC-FT/RP-F8 fraction is shown in Figure 4A. The major peaks in the 20 to 30 min range of retention time (t_R_) in this fraction were particularly interesting since they were regarded as abundant peptides. As shown in Table 1, 14 peptides were simultaneously identified in HILIC-FT/RP-F8 fraction using LC-MS/MS coupled with de novo sequencing. Among these identified peptides, ALVY (*m*/*z* 465 at t_R_ = 24.88 min) showed the highest peak intensity in the LC-MS/MS chromatogram, which implied it was the most abundant peptide in the HILIC-FT/RP-F8 fraction (Figure 4A). To find out the most potential ACE inhibitory peptides from these 14 identified peptides, their bioactivities were predicted using the bioinformatics tools BIOPEP database. The prediction of the ACE inhibitory activity of proteins followed by BIOPEP was described as exemplified by the peptides from amaranthus hydrolysate reported previously [26]. Full or partial peptide sequences can be checked in the BIOPEP database, which gives a relevant and reliable manner to either predict or suspect possible bioactivities of identified peptides. The peptides with the top four high BIOPEP scores, ALVY (0.078), LLVY (0.078), LSTSTDVR (0.064), and LLAPHY (0.055), were regarded as ACE inhibitory peptide candidates. The N-terminal and C-terminal residues in a peptide sequence play a crucial role in dominating the peptide’s bioactivity [34]. The presence of the N-terminal G, I, L, and V in a peptide chain is preferable for ACE inhibition, while P, Y, R/K, and W residues are the favored C-terminal amino acids [35,36]. According to the literature, potent ACE inhibitory peptides contained hydrophobic amino acid residues (V, L, P, G, and A) at the N-terminus, and aromatic residues (Y, F) at the C-terminus accompanied with the decreased side-chain size of a residue in an adjacent position [34]. The N- and C-termini of the peptides ALVY, LLVY, and LLAPHY identified in this study perfectly matched the characteristics of potent ACE inhibitory peptides reported previously. The characteristics of LSTSTDVR were also consistent with the structure-activity correlation proposed by Fujita et al. [37] in which the positive charge on the C-terminal basic residue or basic residue (R/K) contributed substantially to the inhibitory potency. The MS/MS spectra of peptides ALVY (*m*/*z* 465), LLVY (*m*/*z* 507), LLAPHY (*m*/*z* 357), and LSTSTDVR (*m*/*z* 439) are shown in Figure 4B–E, respectively. These peptides’ identities and ACE inhibitory activities were further confirmed using synthetic peptides.

### 2.5. IC_50_ Values of Four Peptide Candidates against ACE

The ACE inhibitory activities of four peptides with high BIOPEP scores were further evaluated by the in vitro ACE inhibitory assay using the corresponding synthetic peptides. The IC_50_ values were determined as 7.03 ± 0.09, 87.76 ± 3.37, 15.91 ± 0.52, and >166.67 µM for ALVY, LLVY, LLAPHY, and LSTSTDVR, respectively (Figure 5). The peptide ALVY had the highest ACE inhibitory activity among four peptide candidates. Besides, ALVY was also the most abundant peptide in the fraction of HILIC-FT/RP-F8, which implied that it is the major peptide that contributed to the strong ACE inhibitory activity in this fraction as well as in the GSPs hydrolysate. To the best of our knowledge, this study is the first report that introduces novel ACE inhibitory peptides from GSPs hydrolysate. Compared to other peptides derived from other food-sourced proteins reported previously, ALVY (IC_50_ = 7.03 ± 0.09 µM) showed a remarkable ACE inhibitory activity superior than those generated from other plant seeds, such as *Cassia obtusifolia* seeds (FHAPWK, IC_50_ = 16.8 μM) [38], barley (FQLPKF, GFPTLKIF, NFLARF, and ALRYFM IC_50_ values ranged from 28.2 to 200 μM) [39], bitter melon seeds (VSGAGRY, IC_50_ = 8.6 μM) [40], hemp seeds (WVYY, WYT, SVYT, and IPAGV, IC_50_ values ranged from 27 to 574 μM) [41], and wheat germ (QV, NPPSV, and VEW, IC_50_ values ranged from 26.8 to 115.2 μM) [42].

### 2.6. Molecular Docking of Synthetic Peptides Sequence at the ACE Binding Site

The molecular docking between the ACE binding site and the four small peptides were simulated using a computational approach that presents the ligand-receptor complex structures through the use of a scoring function. The Discovery Studio software was used to perform docking between the synthetic peptide sequence and ACE binding site. The molecular docking supported that the effectiveness of the peptides’ ACE inhibitory activity was due to the creation of H-bonds with the residues within or outside of the active site of ACE, blocking the active site or distorting the catalysis configuration, respectively. The simulated interactions of ALVY, LLVY, LSTSTDVR, and LLAPHY with the residues surrounding the ACE active site are shown in Figure 6. The peptide ALVY interacted with ACE through three H-bonds with the residues His353, His513, and Tyr523 of ACE (Figure 6A); LLVY and LSTSTDVR had only one H-bond at Tyr523 and Glu384, respectively (Figure 6B,C); peptide LLAPHY formed three H-bonds at His353, Tyr523, and Ser213 of ACE (Figure 6D). Moreover, ALVY exhibited four van der Waals interactions with ACE at His353, Tyr523, Arg522, and Ser386; LLVY had two van der Waals at Ala356 and Tyr523; LSTSTDVR connected four van der Waals at Arg522, His513, Glu384, and His383; LLAPHY had three van der Waal interactions at His353, Tyr523, and Arg124 with ACE. According to the literature, there are three pockets (S1, S2, and S1′) and one zinc-binding domain within the ACE active site: the S1 pocket has key residues, such as Ala354, and Glu384; the S2 pocket includes Tyr523, Gln281, His353, Lys511, His513, and Tyr520; the S1′ pocket has one key residue Glu162, while the zinc binding motif contained His382, His387, and Glu411 residues [26,43]. The interactions between the four ACE inhibitory peptides and ACE are summarized in Table 2. The docked conformations of the peptide ligands were listed by the increasing energy order: LLVY, ALVY, LLAPHY, and LSTSTDVR.

### 2.7. Kinetics of ACE Inhibition with ALVY Sequence

The IC_50_ value of ALVY was previously determined to be 7.03 ± 0.09 µM for ACE inhibition. The ACE inhibition kinetic properties were carefully examined in order to discover the underlying mechanisms that allow peptides to serve as potential antihypertensive agents. In this study, Lineweaver-Burk plots were used to evaluate the ACE inhibition mode of the ALVY sequence. In Figure 7, it is apparent that ALVY was indicated as a competitive inhibitor of ACE. The result suggests that ALVY can bind to the active site of ACE, thus blocking the binding of ACE to the substrate and inhibiting the activity of ACE. The Vmax was not significantly affected by the concentration; however, Km increased with the increasing peptide concentration. This result was perfectly supported by the molecular docking of ALVY in the active site of ACE. The competitive inhibitors are able to enter the ACE protein molecule, interact with the active sites, and prevent substrate binding [44]. In explaining the competitive inhibition of the inhibitors, the C-terminal peptide amino acids possibly will interact with the substrates S1, S1′, and S2′ at the active site of ACE. The zinc ion of ACE is correctly positioned between S2 and S1′ to contribute in the hydrolytic cleavage of the substrate peptide bond, causing the release of the dipeptide product. ACE active sites S1, S1′, and S2′ have high attractions for the hydrophobic side chains of tryptophan, alanine, and proline [45,46].

## 3. Materials and Methods

### 3.1. Materials

Gac seeds were collected in Pingtung, Taiwan. Acetone, acetonitrile (ACN), and hexane were purchased from Fisher Scientific (Seoul, Korea); trichloroacetic acid (TCA) and chymotrypsin, formic acid (FA), pepsin, sodium dodecyl sulfate (SDS), trypsin, Hippuryl-L-Histidyl-L-Leucine (HHL), dimetylformamide (DMF), triisopropylsilane (TIS), N,N’-diisopropylacarbodiimide (DIC), and ethyl ether were purchased from Sigma-Aldrich (St Louis, MO, USA); trifluoroacetic acid (TFA), 4-methylmorpholine (NMM), N-ethyl-19-diisopropylamine (DIPEA), and O-(1H-benzotriazol-1-yl)-N.N.N’,N’ Tetramethyl-hexafluorophosphate (HOBt) were purchased from Alfa Aesar (Heysham, Lancashire, UK); ultra-filtration membranes (3 kDa MWCO) were obtained from Millipore (Bedford, MA, USA); and methyl alcohol from Macron Fine Chemicals^TM^ (Radnor, PE, USA). Oxyma pure and Fmoc-amino acids were obtained from Novabiochem (Billerica, MA, USA). Methylene chloride (DCM) was purchased from Duksan (Sinworo, Korea). Other chemicals used in this experiment were of analytical grade. The water used in this study was generated using the PURELAB^®^ water purification system from ELGA LabWater (Lane End, High Wycombe, UK).

### 3.2. Preparation and Extraction of Gac Seed Protein Powder

Fresh gac seeds were washed by hand and tray-dried (hot air tray dried HTD-7, Jaw Chuange Machinery Co., Ltd., Taoyuan, Taiwan) for 36 h at 40 °C to reduce the moisture. The dried gac seeds were pulverized to a homogenous powder in a laboratory grinder (grinder RTRT, Taichung, Taiwan). The dried gac seed powder was defatted in triplicate with 1:5 *w*/*v* of hexane at room temperature for 30 min. The residue (defatted meal) was collected and air-dried in a fume hood for 24 h. The defatted gac seed powder was stored in a desiccator at room temperature until used. Sixty milligrams of defatted gac seed powder were extracted in triplicate with 1% sodium dodecyl sulfate (SDS) (6 mL) at 0 °C in a sonicator (Branson Digital Sonifier^®^, Hampton, Rockingham, NH, USA) at 30% potential for 3 min, then centrifuged (Hitachi himac CT15RE, Hitachi Koki Co., Ltd., Tokyo, Japan) at room temperature, 15,000× *g* for 10 min. The supernatants were collected, concentrated by a Freeze Dryer (Panchum Scientific crop, Kaohsiung, Taiwan), and then proteins were precipitated using 10% trichloroacetic acid (TCA) in acetone at −20 °C for 24 h. The proteins were centrifuged at room temperature, 15,000× *g* for 10 min and washed in triplicate with 6 mL of acetone. The protein pellet was washed with 6 mL of deionized water, centrifuged at room temperature, and freeze-dried. The gac seed proteins were stored at −20 °C for further experiments.

### 3.3. Gac Seed Protein Hydrolysis

Crude protein extract from gac seeds was hydrolyzed (in triplicate) using the combination of three different proteases: pepsin, trypsin, and chymotrypsin. The crude protein pellet (10 mg) was suspended in 35 mM sodium chloride (NaCl) and adjusted to pH 2 using 4 M HCl. Pepsin (0.2 mg, 250 units/mg) was added and the reaction mixture was incubated in an incubator (YIH DER, LM-570D, Yihder Technology Co., Ltd., New Taipei, Taiwan) at 37 °C for 24 h. After, the protein mixture was adjusted to pH 8 to inactivate pepsin with 5 M NaOH solution. Subsequently, 0.2 mg of trypsin (250 units/mg) and 0.2 mg of chymotrypsin (40 units/mg) were added to the protein mixture and the mixture was incubated at 37 °C for 24 h. The small peptides were then isolated using ultrafiltration membrane (Amicon^®^ Ultra-0.5, Merck Millipore, Darmstadt, Germany) with a molecular weight cutoff at 3 kDa and centrifuged 10,000× *g* at 10 °C for 10 min. The filtrate (MW < 3 kDa) was then freeze-dried and kept at −20 °C for further assay or analysis.

### 3.4. C_18_ Solid-Phase Extraction (SPE) for Desalting of GSPs Hydrolysate

The peptides were dissolved in deionized water and desalted using solid-phase extraction (SPE) on HyperSep^TM^ C_18_ Columns (Thermo Scientific, TN, USA). SPE columns were activated prior to use with 1 mL methanol and 1 mL 95% acetonitrile + 0.1% formic acid to wash the resin in the column. In total, 1 mL of 5% acetonitrile containing 0.1% formic acid was added to equilibrate the column. Then, the peptide samples (1 mL) were loaded onto the column and washed three times with 1 mL of 5% acetonitrile containing 0.1% formic acid. Finally, the peptides were eluted using 1 mL of 50% acetonitrile containing 0.1% formic acid and the remaining peptides were subsequently eluted using 1 mL of 80% acetonitrile containing 0.1% formic acid. The eluted fractions were combined, freeze-dried, and stored at −20 °C prior to further experiments.

### 3.5. Fractionation of GSPs Hydrolysate Using Hydrophilic Interaction Liquid Chromatography (HILIC)

The peptides were dissolved in 95% acetonitrile containing 0.1% formic acid and purified using hydrophilic interaction liquid chromatography (HILIC) on columns of ZIC^®^—HILIC SPE (SeQuant, Umea, Sweden). HILIC columns were activated prior to use with 1 mL of methanol followed by 1 mL of 5% acetonitrile containing 0.1% formic acid. Then, the column was equilibrated using 95% acetonitrile containing 0.1% formic acid. After, the peptide sample was loaded onto the column, and the flow-through fraction was collected. The column was added with 1 mL of 100%, 90%, 80%, 70%, 60%, 50%, and 0% acetonitrile containing 0.1% formic acid, sequentially. Each eluted fraction was collected, freeze-dried, and stored at −20 °C prior to further experiments.

### 3.6. Fractionation of Gac Seed Peptides Using RP-HPLC

The in vitro ACE inhibitory assay indicated that the HILIC flow-through fraction showed the highest ACE inhibition among eight fractions. Therefore, the flow-through fraction was further separated using reversed-phase high-performance liquid chromatography (RP-HPLC). The sample (1 mg) dissolved in 5% acetonitrile containing 0.1% formic acid was injected and separated by HPLC (Hitachi Chromaster, Tokyo, Japan) with a C_18_ column (4.6 mm × 250 mm; particle size 5 µm)(Phenomenex, Torrance, CA, USA). The mobile phase was arranged using solvent A (5% acetonitrile containing 0.1% trifluoroacetic acid) and solvent B (95% acetonitrile containing 0.1% trifluoroacetic acid) at 1 mL/min. The fractionation was performed using a gradient as follows: 100% A from 0 min; 100–80% A from 0–45 min; 80–20% A from 45–85 min, and 20% A from 85–90 min then returning to initial conditions. The separation was monitored at UV 214 nm. Peptide fractions were collected peak by peak. All peptide fractions were freeze-dried and stored at −20 °C prior to further experiments.

### 3.7. Angiotensin I-Converting Enzyme (ACE) Inhibitory Activity Assay and IC_50_ Determination

The method to determine the ACE inhibitory activity of gac seeds peptides hydrolysate was adapted according to a previous report with slight modification [47]. Briefly, the sample dissolved in ddH_2_O (10 µL) was added to 2.5 mM hippuryl-L-histidyl-L-leucine (30 µL). Borate buffer (10 µL) was used as blank solution. In total, 50 µM Captopril (10 µL) was used as positive control. The samples, blank, and positive control were incubated at 37 °C for 5 min. Then, 0.05 mU/µL of ACE (20 µL) was added into each solution. The reaction mixture was incubated at 37 °C for 30 min and mixed in a shaker incubator at 200 rpm for 30 min. The reaction was stopped by adding 1 N HCl (60 µL). After, 0.25 mg/mL ferulic acid (10 µL) was added as an internal standard. Samples were centrifuged (Hitachi Koki Co., Ltd., Tokyo, Japan) at 15,000 rpm for 2 min. The samples were separated by RP-HPLC using a C_18_ column (4.6 mm × 250 mm, particle size 5 µm) (Macherey-Nagel, Duren, Germany). Samples were selected using an isocratic gradient of 95% solvent A and 15% solvent B at a flow rate of 1 mL/min. Solvent A was prepared by 5% acetonitrile + 0.1% trifluoroacetic acid and solvent B was prepared by 95% acetonitrile + 0.1% trifluoroacetic acid. Absorbance was measured using a UV detector at 228 nm. ACE inhibitory activity (%) was examined in triplicate and calculated according to the following equation:ACE-inhibitory activity (%) = [1 − (ΔA_inhibition_)/(ΔA_control_)] × 100%,
where ΔA_inhibition_ and ΔA_control_ are the peak areas in the sample with or without inhibitor.

The definition of IC_50_ is the peptide concentration required to inhibit 50% of angiotensin-converting enzyme. The IC_50_ was calculated using non-linear regression of the ACE inhibitory activity (%) from nine different concentrations. This assay was repeated three times for each experiment.

### 3.8. Identification of Gac Seed Peptides Sequence Using LC-MS/MS

The peptide sequences in the fraction that showed the most potent ACE inhibitory activity were determined by liquid chromatography-tandem mass spectrometry (LC-MS/MS). Samples dissolved in 5% acetonitrile + 0.1% formic acid were separated using an Ultimate 3000 RSLC system (Dionex, Sunnyvale, CA, USA) coupled with a C_18_ column (Acclaim PepMap RSLC, 75 μm × 150 mm, Thermo Scientific, Waltham, Middlesex, USA) and analyzed with a Thermo Q-Exactive^TM^ mass spectrometer (Thermo Scientific Inc., Waltham, Middlesex, MA, USA). The mobile phase was composed of solution A (0.1% formic acid in water) and solution B (0.1% formic acid in 95% acetonitrile), and the elution gradient included (i) in the first 5.5 min, isocratic at 1% solution B for sample loading, (ii) in the next 39.5 min, linear from 1% to 60% solution B, (iii) in the following 10 min, linear from 60% to 80% solution B, and finally (iv) isocratic at 1% solution B for 10 min. The flow rate in the analytical column was maintained at 250 µL/min. The Q-Exactive^TM^ mass spectrometer was operated in the data-dependent mode to switch automatically between MS and HCD-MS2. Full MS scan was carried out in a 200 and 2000 *m*/*z* range, and the 14 ions of the highest intensities were selected for MS/MS scans. The RAW data was converted to MGF file format followed by automated de novo sequencing using PEAKS Studio X (Bioinformatics Solutions Inc., Waterloo, ON, Canada).

### 3.9. PEAKS Studio X de Novo Sequencing

The LC-MS/MS raw data were analyzed using PEAKS studio X software. The parameter used: (1) Open project structure: added data; (2) Enzyme: GI (combination of three or more enzymes); (3) Type of Enzyme: Unspecific; (4) Instrument: Orbitrap (orbi-trap); (5) Parent Mass error Tolerance: 20 ppm; (6) Fragment Mass Error Tolerance: 0.6 Da; (7) de Novo Score: >70%. The sequence identified through the database was validated by manually matching with the MS/MS spectrum in the raw data. In addition to MS/MS analysis coupled with PEAKS studio, the identified ACE inhibitory peptides were confirmed using the synthetic peptide with corresponding sequences by comparison with their LC retention times, *m*/*z* values, and MS/MS spectra.

### 3.10. Synthesis of Peptides from HILIC-FT/RP 8 Fraction

The peptide sequences identified from the HILIC-FT/RP-8 fraction were uploaded to the BIOPEP database (http://www.uwm.edu.pl/biochemia/index.php/pl/biopep) to simulate their ACE inhibitory activities. The peptides with the top four high scores against ACE in silico predicted by BIOPEP were chosen for further ACE inhibitory assay. To obtain these peptides, solid-phase peptide synthesis was performed using CEM Discover reactor (CEM Microwave Technology Ltd., Buckingham, UK) followed by a solid phase procedure using Fmoc protected amino acids synthesis methods. The Wang resin (0.96 mmole/g) was soaked in 100% DMF for 30 min. Qxyma Pure (0.5 mmloe), DIC (0.5 mmole), DIPEA (1 mmol), and the first amino acid from C-terminal were mixed by a stirrer at room temperature for 8 h. Unreacted hydroxyl groups on resin were blocked using a capping solution (1 mL of 4.5% NMM, 1 mL of acetic anhydride, and 3 mL of DMF). The deprotection of Fmoc at the first amino acid was achieved using 20% piperidine in DMF. The resin was washed with DMF three times after each coupling and five times after deprotection. The next amino acid with Fmoc protection dissolved in the solution composed of 4 mL DMF, 1 mL 4.5% NMM, and HBTU (0.3 mmol) was coupled to the first amino acid. The peptide synthesis was carried out by repeating the deprotection-coupling cycle until the last amino acid was coupled. The resin was washed with DCM three times and the peptide was released from the resin using 3% H_2_O, 3% Tis, and 97% TFA. After washing with TFA two times, the product was precipitated using diethyl ether, centrifuged (Hitachi Koki Co., Ltd., Tokyo, Japan) (8000 rpm for 10 min), freeze-dried (−20 °C), and then the purified peptides were acquired. The synthesized peptides were further purified by HPLC to give purity greater than 95%. The purified synthetic peptides were stored at −20 °C prior to further experiments.

### 3.11. Molecular Docking of Purified Peptides on the ACE Binding Site

The crystal structure of human ACE complexed with the inhibitor lisinopril (PDB: 108a) was derived from the RCSB Protein Data Bank. Before the docking, all water molecular and the inhibitor lisinopril were removed whereas the cofactors Zn^2+^ and chloride ions were retained in the active site of the ACE model. The three-dimensional structure of the purified peptides was drawn using the molecular simulation software Accelry Discovery Studio Visualizer 3.0 (Accelry Software Inc., BIOVIA, San Diego, CA, USA). The docking process was performed, and the energy was minimized by the CHARMM program. The Discovery Studio software scored the docking result according to its own scoring function; based on the scores and the combined free energy of each result, the best matching result was selected.

### 3.12. Kinetics Study of ACE Inhibition

Kinetics of ACE inhibition was determined by a previous method with some modifications [48]. The hydrolysis rate of HHL by ACE over a range of HHL concentrations of 0.005, 0.1, 0.5, 1, and 2 mM was measured while varying the concentrations of ALVY. The modes of ACE inhibition were determined from Lineweaver-Burk plots while inhibition parameters (*V_max_* and *K_m_*) were calculated, respectively, as the Y- and X-axis intercepts of the primary plot.

### 3.13. Statistical Analysis

Experimental results were used for analysis of variance using SPSS version 17. Significant differences between means were determined by Duncan’s multiple-range test (*p* < 0.05).

## 4. Conclusions

The results from this study clearly indicate that gac seed protein hydrolysates have rich ACE-inhibitory activity. The gac seed proteins hydrolysate (<3 kD) demonstrated high ACE inhibitory activity (IC_50_ = 70.0 ± 4 µg/mL). A further fractionation by HILIC indicated that the FT fraction has a higher ACE inhibitory activity than the other fractions. Regarding RP-HPLC fractionation, HILIC-FT/RP-F8 fraction has potent ACE inhibitory activity and 14 peptides were identified from this fraction using LC-MS/MS analysis coupled with de novo sequencing. The peptide ALVY derived from GSPs hydrolysate exhibited superior ACE inhibitory activity, with an IC_50_ value of 7.03 ± 0.09 µM, among the peptides, which occurred competitively according to the Lineweaver-Burk plot. Molecular docking results suggested that the ACE inhibition of ALVY was mainly attributed to forming a strong hydrogen bond with the S1 pocket (Tyr523) and the S2 pocket (His353 and His513). This finding indicated that antihypertensive peptides derived from gac seed protein hydrolysates may be a potential resource of functional peptides for drug applications.

## Figures and Tables

**Figure 1 molecules-25-04635-f001:**
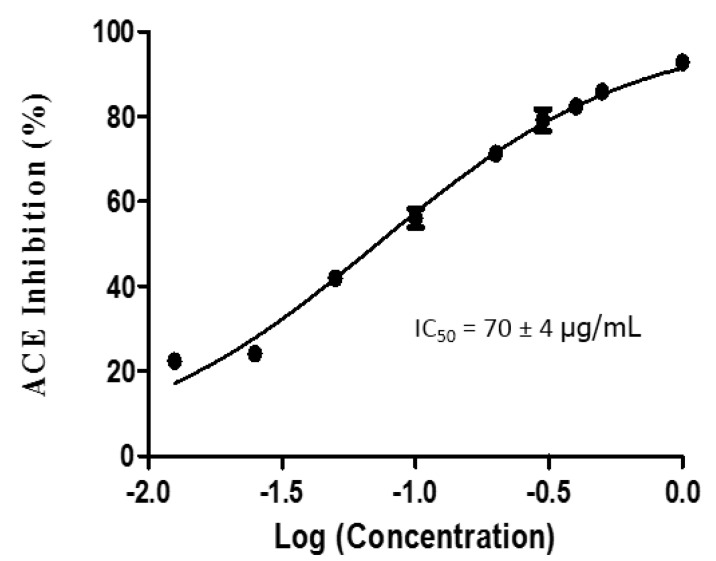
IC_50_ value of gac seed proteins (GSPs) hydrolysate (<3 kD).

**Figure 2 molecules-25-04635-f002:**
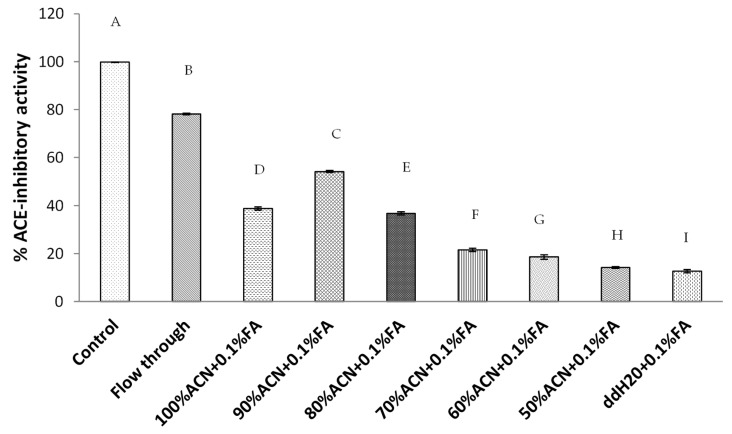
ACE inhibitory activities of different HILIC fractions derived from GSPs hydrolysate (<3 kD). These HILIC fractions include flow-through (FT), 100%, 90%, 80%, 70%, 60%, and 50% ACN containing 0.1% FA. Means denoted by a different letter indicate significant differences between treatments (*p* < 0.05).

**Figure 3 molecules-25-04635-f003:**
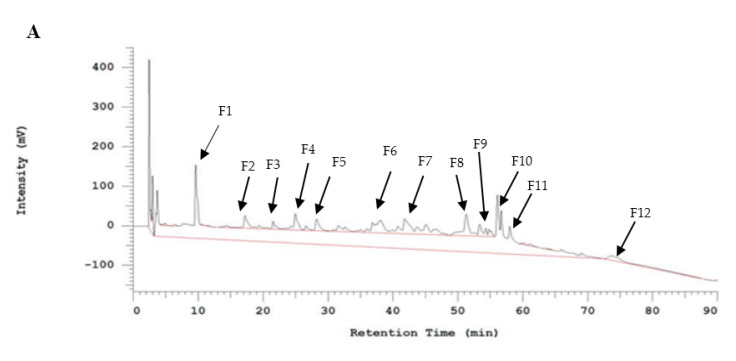
(**A**) RP-HPLC chromatogram of HILIC FT fraction, (**B**) ACE inhibitory activities of 12 fractions of HILIC FT separated using RP-HPLC. Means denoted by a different letter indicate significant differences between treatments (*p* < 0.05).

**Figure 4 molecules-25-04635-f004:**
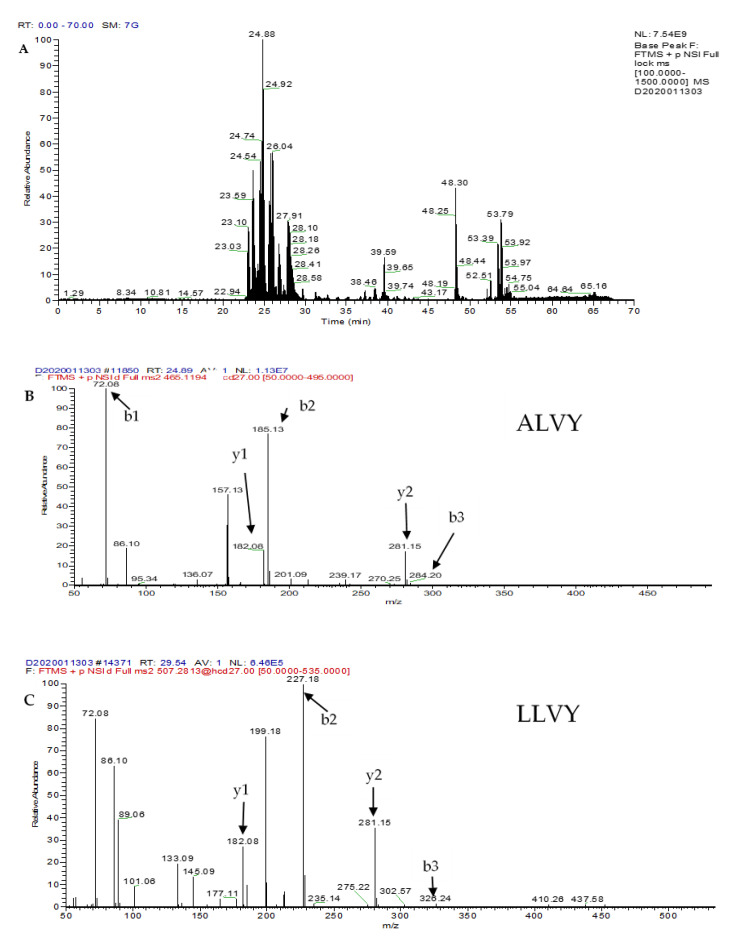
LC-MS chromatogram of HILIC-FT/RP-F8: (**A**) Full LC-MS chromatogram, (**B**) ALVY; MS/MS spectrum at *m*/*z* 465, (**C**) LLVY; MS/MS spectrum at *m*/*z* 507, (**D**) LLAPHY; MS/MS spectrum at *m*/*z* 357.43066 and (**E**) LSTSTDVR; MS/MS spectrum at *m*/z 439. All MS/MS spectra were performed under positive mode.

**Figure 5 molecules-25-04635-f005:**
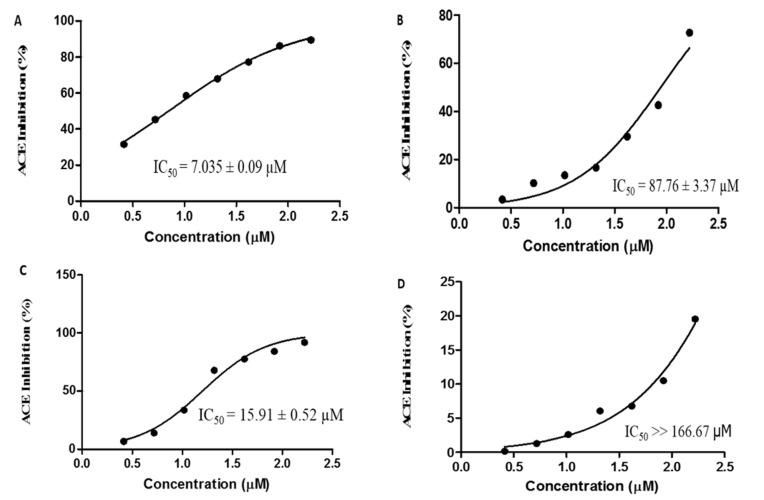
IC_50_ values of synthetic peptides: (**A**) ALVY, (**B**) LLVY, (**C**) LLAPHY, and (**D**) LSTSTDVR peptides.

**Figure 6 molecules-25-04635-f006:**
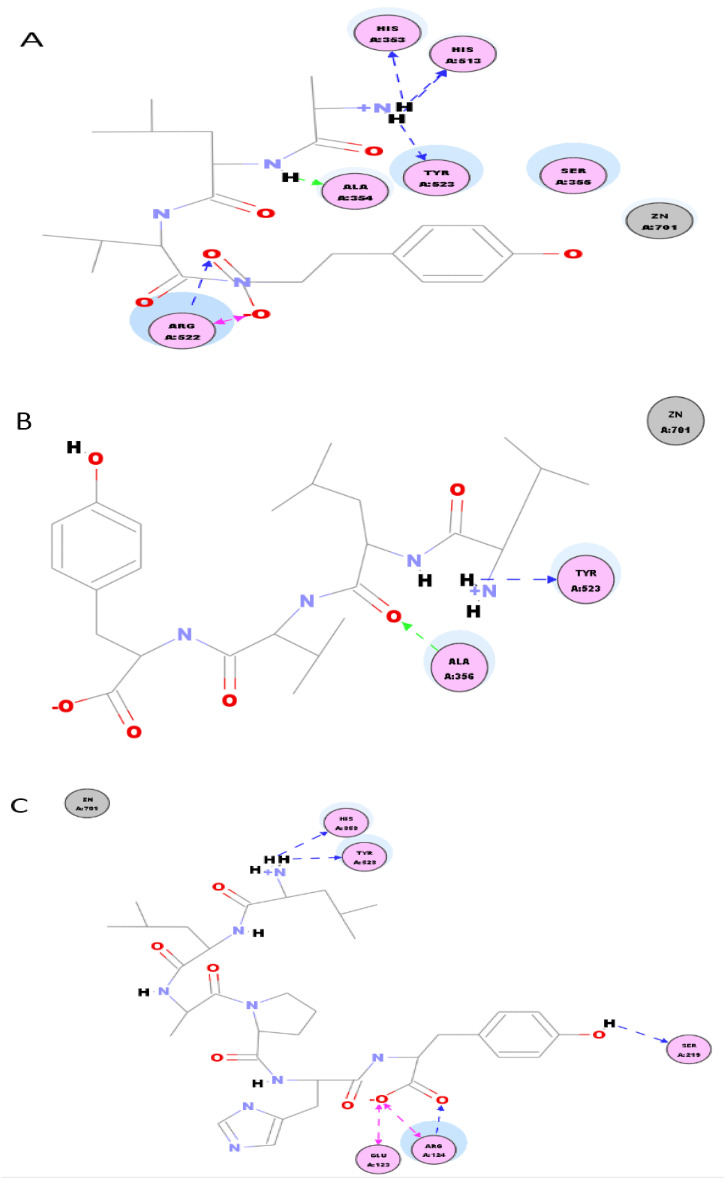
The molecular interactions of four peptides (identified from HILIC-FT/RP-F8 fraction) with ACE pockets: (**A**) ALVY, (**B**) LLVY, (**C**) LLAPHY, and (**D**) LSTSTDVR peptides.

**Figure 7 molecules-25-04635-f007:**
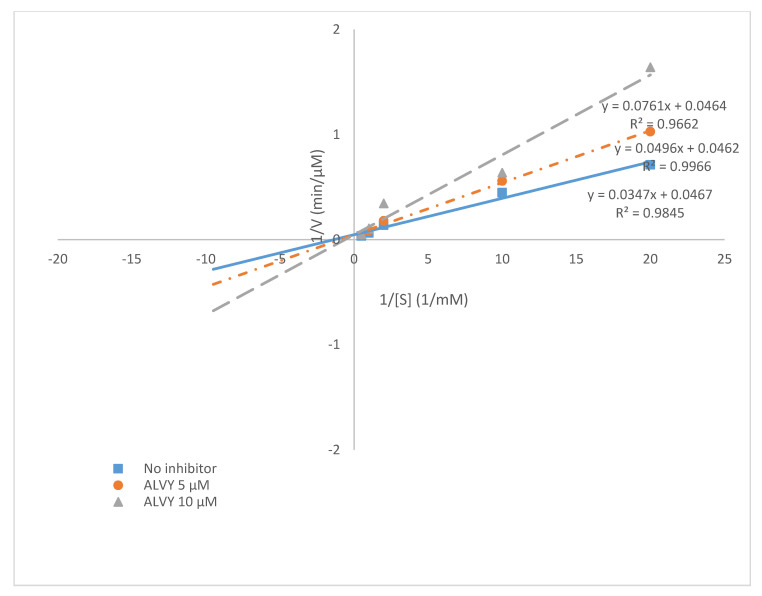
Lineweaver-Burk plot of ACE inhibitory by ALVY.

**Table 1 molecules-25-04635-t001:** Identification and bioactivity of peptides from flow through RP-HPLC fraction 8.

Sequence	Mass (*m*/*z*)	RT (min)	Score	Biopep ^(a)^	Bioactivities
LPGY	449.2394	23.47	93	0.044	ACE-inhibitory
LLPGY	562.3236	28.20	89	0.039	ACE-inhibitory
LLVY	507.3176	29.41	88	0.078	ACE-inhibitory
ALVY	465.2692	24.88	88	0.078	ACE-inhibitory
VPAVL	498.3264	26.31	87	0.028	ACE-inhibitory
LLAPHY	357.2030	22.74	86	0.055	ACE-inhibitory
LGVF	435.2602	26.25	82	0.028	ACE-inhibitory
ALLY	479.2861	25.43	80	0.013	ACE-inhibitory
LLNY	522.2913	25.67	79	0.010	ACE-inhibitory
ALEAY	566.2821	21.59	78	0.010	ACE-inhibitory
VLPPLE	667.4026	29.33	78	0.018	ACE-inhibitory
AEVF	465.2340	24.76	78	0.027	ACE-inhibitory
LPPPPGL	593.3659	26.27	76	0.031	ACE-inhibitory
LSTSTDVR	439.7324	20.85	74	0.064	ACE-inhibitory

^(**a**)^ Biopep database; http://www.uwm.edu.pl/biochemia/index.php/pl/biopep.

**Table 2 molecules-25-04635-t002:** Molecular docking of 4 peptides (identified from HILIC-FT/RP-F8 fraction) with ACE pockets.

Sequence	ACE Pockets	CDOCKER_INTERACTION_ENERGY (KJ/mol)
**ALVY**	His353, His513, Tyr523, Ala354, Arg522	−69.2269
**LLVY**	Ala356, Tyr523	−65.9744
**LSTSTDVR**	Arg124, Arg122, His513, Glu384, His383	−105.5880
**LLAPHY**	His353, Ser219, Glu123, Arg124	−86.8579

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
