# Peer review of "ACE Inhibitory Activity and Molecular Docking of Gac Seed Protein Hydrolysate Purified by HILIC and RP-HPLC"

_molecules, 2020, doi:10.3390/molecules25204635_

Round 1

Reviewer 1 Report

The manuscript by Ngamsuk et al. was reported novel ACE inhibitory peptides from Gac seed protein. They prepared protein hydrolysate by proteinases, determined several novel ACE inhibitory peptides by HILIC and RP-HPLC and characterized the ACE inhibitory function by using synthetic peptide and molecular docking method. The study was well organized and meets for the publication of Molecules with some modifications.

Authors should describe the source of proteins in ACE inhibitory peptides. The information is important for the prediction of the peptide from relative seed. In addition, the information increase the reliability of the identified peptides.

Line 107, Please described the amount of dried weight of the started sample and yield of protein hydrolysate.

Line 186, Authors should add units of ACE.

Line 314, What means the “minus” of ACE inhibitory activity? Does this mean that these peptide fractions increase ACE activity? Authors should clarify this point.

Table 1. Authors should mention red character.

Figure 4, red lines between figure A and B should be deleted.

Line 417, Does the peptide ALLVY means ALVY?

Line 417, The energy order of peptides differ from Table 2. Authors shows the order :ALLVY, LLVY, LSTSTDVR, and LLAPHY. But Table 2 indicates LLVY showed lowest (-65.9744) ALVY (-69.2269), LSTSTDVR (-105.5880) and LLAPHY (-86.8579).

Figure 6. More high resolution file will need. The letter in the Figure cannot see.

Authors clarified the relationship between IC50 and docking score of four peptides.

Please confirm authors name of Reference 33: Fujita and Yokoyama.

Author Response

Author’s Reply to the Review Report (Reviewer 1)

Comment 1: Line 107, Please described the amount of dried weight of the started sample and yield of protein hydrolysate.

Response: Thanks for Reviewer’s comment. The amount of dried weight of the started sample did not do in this experiment. The amount of dried weight started at defatted gac seed power 60 mg for extracted protein and the yield of protein was 7 mg; calculated according to the following equation:

%Yield = (Output/Input)*100 = (7 mg/ 60 mg)*100 = 11.66%  

Comment 2: Line 186, Authors should add units of ACE.

Response: Thanks for Reviewer’s comment. The ACE inhibitory activity is based on the relative activity compared with the control (without inhibitor). Therefore, ACE inhibitory activity was calculated as %.

Comment 3: Line 314, What means the “minus” of ACE inhibitory activity? Does this mean that these peptide fractions increase ACE activity? Authors should clarify this point.

Response: Thanks for Reviewer’s comment. The line 314 has been change to line 320. The small minus ACE inhibitory activity was due to the experimental bias at very low ACE inhibitory activity. To avoid misunderstanding, the minus ACE inhibitory activity was revised to ~ 0%.

Comment 4: Table 1. Authors should mention red character.

Response: Thanks for Reviewer’s comment. The red character pointed out the peptides with top four high BIOPEP scores. To avoid readers’ misunderstanding, the red color was removed.

Comment 5: Figure 4, red lines between figure A and B should be deleted.

Response: Thanks for Reviewer’s suggestion.

Comment 6: Line 417, Does the peptide ALLVY means ALVY?

Response: Thanks for Reviewer’s correction.  The line 417 has been change to line 438. It has been revised. 

Comment 7: Line 417, The energy order of peptides differ from Table 2. Authors shows the orderALLVY, LLVY, LSTSTDVR, and LLAPHY. But Table 2 indicates LLVY showed lowest (-65.9744) ALVY (-69.2269), LSTSTDVR (-105.5880) and LLAPHY (-86.8579).

Response: Thanks for Reviewer’s correction. The line 417 has been change to line 438. It has been revised from “ALLVY, LLVY, LSTSTDVR and LLAPHY” to “LLVY, ALVY, LLAPHY and LSTSTDVR”.

Comment 8: Figure 6. More high resolution file will need. The letter in the Figure cannot see.

Response: Thanks for Reviewer’s suggestion. It has been replaced by a higher resolution figure.

Comment 9: Authors clarified the relationship between IC50 and docking score of four peptides.

Response:  Thanks for Reviewer’s suggestion. The CDOCKER energy between inhibitory peptide and enzyme relies on many factors, such as the number of interactions (eg. H-bonding, pi-pi interaction, charge-charge interaction, etc.), the interaction distances, number of interacting residues, and so on. Therefore the CDOCKER energy alone cannot completely correlate their inhibitory activities (IC50). Instead, the interaction with key residues in active site is crucial to dominant the inhibitory activity. In this case, the ability of peptides to develop numerous hydrogen bond interactions with key residues in ACE may be a major factor in the ACE inhibitory activity of the peptide and stabilization of the ACE peptide complex structure. IC50 is correlated to the number of hydrogen bonds formed. ALVY revealed the lowest IC50 (7.035 ± 0.09 µM) of -69.2269 KJ/mol and formed 3 H-bonds, followed by LLVY with an IC50 (87.76 ± 3.37 µM) of -65.9744 KJ/mol and formed 1 H-bond, followed by LSTSTDVR with an IC50 (1028 ± 79.88 µM) of -105.5880 KJ/mol and formed 1 H-bond and LLAPHY with an IC50 (15.91 ± 0.52 µM) of -86.8579 KJ/mol. Thus, it appears that the number of hydrogen bonds and ACE key residues involved play a prominent role in the ACE-inhibitory capacity of the peptides.

Comment 10: Please confirm authors name of Reference 33: Fujita and Yokoyama.

Response:  Thanks for Reviewer’s comment. The reference 33 has been change to the reference 39. The reference has been replaced with “Fujita, H.; Yokoyama, K.; Yasumoto, R.; Yoshikawa, M. Anti-hypertensive effect of thermolysin digest of dried bonito in spontaneously hypertensive rat (SHR). Clin. Exp. Pharmacol. Physiol. 1995, 22(s1),S304-S305”

Author’s Reply to the Review Report (Reviewer 1)

Comment 1: Line 107, Please described the amount of dried weight of the started sample and yield of protein hydrolysate.

Response: Thanks for Reviewer’s comment. The amount of dried weight of the started sample did not do in this experiment. The amount of dried weight started at defatted gac seed power 60 mg for extracted protein and the yield of protein was 7 mg; calculated according to the following equation:

%Yield = (Output/Input)*100 = (7 mg/ 60 mg)*100 = 11.66%  

Comment 2: Line 186, Authors should add units of ACE.

Response: Thanks for Reviewer’s comment. The ACE inhibitory activity is based on the relative activity compared with the control (without inhibitor). Therefore, ACE inhibitory activity was calculated as %.

Comment 3: Line 314, What means the “minus” of ACE inhibitory activity? Does this mean that these peptide fractions increase ACE activity? Authors should clarify this point.

Response: Thanks for Reviewer’s comment. The line 314 has been change to line 320. The small minus ACE inhibitory activity was due to the experimental bias at very low ACE inhibitory activity. To avoid misunderstanding, the minus ACE inhibitory activity was revised to ~ 0%.

Comment 4: Table 1. Authors should mention red character.

Response: Thanks for Reviewer’s comment. The red character pointed out the peptides with top four high BIOPEP scores. To avoid readers’ misunderstanding, the red color was removed.

Comment 5: Figure 4, red lines between figure A and B should be deleted.

Response: Thanks for Reviewer’s suggestion.

Comment 6: Line 417, Does the peptide ALLVY means ALVY?

Response: Thanks for Reviewer’s correction.  The line 417 has been change to line 438. It has been revised. 

Comment 7: Line 417, The energy order of peptides differ from Table 2. Authors shows the orderALLVY, LLVY, LSTSTDVR, and LLAPHY. But Table 2 indicates LLVY showed lowest (-65.9744) ALVY (-69.2269), LSTSTDVR (-105.5880) and LLAPHY (-86.8579).

Response: Thanks for Reviewer’s correction. The line 417 has been change to line 438. It has been revised from “ALLVY, LLVY, LSTSTDVR and LLAPHY” to “LLVY, ALVY, LLAPHY and LSTSTDVR”.

Comment 8: Figure 6. More high resolution file will need. The letter in the Figure cannot see.

Response: Thanks for Reviewer’s suggestion. It has been replaced by a higher resolution figure.

Comment 9: Authors clarified the relationship between IC50 and docking score of four peptides.

Response:  Thanks for Reviewer’s suggestion. The CDOCKER energy between inhibitory peptide and enzyme relies on many factors, such as the number of interactions (eg. H-bonding, pi-pi interaction, charge-charge interaction, etc.), the interaction distances, number of interacting residues, and so on. Therefore the CDOCKER energy alone cannot completely correlate their inhibitory activities (IC50). Instead, the interaction with key residues in active site is crucial to dominant the inhibitory activity. In this case, the ability of peptides to develop numerous hydrogen bond interactions with key residues in ACE may be a major factor in the ACE inhibitory activity of the peptide and stabilization of the ACE peptide complex structure. IC50 is correlated to the number of hydrogen bonds formed. ALVY revealed the lowest IC50 (7.035 ± 0.09 µM) of -69.2269 KJ/mol and formed 3 H-bonds, followed by LLVY with an IC50 (87.76 ± 3.37 µM) of -65.9744 KJ/mol and formed 1 H-bond, followed by LSTSTDVR with an IC50 (1028 ± 79.88 µM) of -105.5880 KJ/mol and formed 1 H-bond and LLAPHY with an IC50 (15.91 ± 0.52 µM) of -86.8579 KJ/mol. Thus, it appears that the number of hydrogen bonds and ACE key residues involved play a prominent role in the ACE-inhibitory capacity of the peptides.

Comment 10: Please confirm authors name of Reference 33: Fujita and Yokoyama.

Response:  Thanks for Reviewer’s comment. The reference 33 has been change to the reference 39. The reference has been replaced with “Fujita, H.; Yokoyama, K.; Yasumoto, R.; Yoshikawa, M. Anti-hypertensive effect of thermolysin digest of dried bonito in spontaneously hypertensive rat (SHR). Clin. Exp. Pharmacol. Physiol. 1995, 22(s1),S304-S305”

"Please see the attachment" we updated the manuscript.

best regards

Samuchaya Ngamsuk

Reviewer 2 Report

Review Comments – molecules-936150

Title of Manuscript: ACE-inhibitory Activity and Molecular Docking of Gac Seed Protein Hydrolysate Purified by HILIC and RP-HPLC

Authors: Samuchaya Ngamsuk *, Tzou-Chi Huang, Jue-Liang Hsu *

General comment

This is an interesting report for ACE inhibitory peptides derived from gac seed protein hydrolysate. The primary aim of this study is effective utilization of gac seeds that have long been considered waste products in gac industry. It deserves publication with some revisions.

Authors should show or discuss the original protein of ACE inhibitory peptides.

Authors descrive that gac seeds have been used as a treatment of certain diseases (diabetes, eye disorders, fluxes, liver spleen disorders, haemorrhoids, wounds, bruises, boils, sores, scrofula, tinea, swelling, and pus) in traditional Chinese medicine. So, they should discuss that the relationship between the diseases and ACE inhibitory activity in this study.

Authors ought to add the references as follows:

“In vitro and in silico approaches to generating and identifying angiotensin-converting
enzyme I inhibitory peptides from green macroalga Ulva lactuca” Marine Drugs,
doi. 10.1111/1750-3841.15115 (2020).

In silico analysis of relationship between proteins from plastid genome of red alga Palmaria sp. (Japan) and angiotensin I converting enzyme inhibitory peptides” Marine Drugs, doi.org/10.3390/md17030190 (2019).

In silico analysis of ACE inhibitory peptides from chloroplast proteins of red alga Grateloupia asiatica” Marine Biotechnology, doi.org/10.1007/s10126-020-09959-2 (2020).

Author Response

Author’s Reply to the Review Report (Reviewer 2)

Comment 1: Authors should show or discuss the original protein of ACE inhibitory peptides.

Response:  Thanks for Reviewer’s kind suggestion. Since the genome of Gac (Momoridica cochinchinensis Spreng.) has not been decoded completely, the protein database was not sufficient for comprehensive database-assisted peptide identification. Therefore, the identification of the original protein is far from straightforward. But the peptide identification can be readily achieved using LC-MS/MS analysis and de novo sequencing. The discussion was included in Section 3.4 in the revised manuscript.

Comment 2: Authors descrive that gac seeds have been used as a treatment of certain diseases (diabetes, eye disorders, fluxes, liver spleen disorders, haemorrhoids, wounds, bruises, boils, sores, scrofula, tinea, swelling, and pus) in traditional Chinese medicine. So, they should discuss that the relationship between the diseases and ACE inhibitory activity in this study.

Response: Thanks for Reviewer’s kind suggestion. Indeed, there is no direct relationship between ACE inhibitory activity and the above-mentioned diseases. To explore more health benefits of gac seeds, this study was aimed to investigate the antihypertensive peptides from enzymatic hydrolysate of gac seeds. We add this description in the revised manuscript.

Comment 3: Authors ought to add the references as follows:

“In vitro and in silico approaches to generating and identifying angiotensin-converting 
enzyme I inhibitory peptides from green macroalga Ulva lactuca” Marine Drugs, 
doi. 10.1111/1750-3841.15115 (2020).

In silico analysis of relationship between proteins from plastid genome of red alga Palmaria sp. (Japan) and angiotensin I converting enzyme inhibitory peptides” Marine Drugs, doi.org/10.3390/md17030190 (2019).

In silico analysis of ACE inhibitory peptides from chloroplast proteins of red alga Grateloupia asiatica” Marine Biotechnology, doi.org/10.1007/s10126-020-09959-2 (2020).

Response: Thanks for Reviewer’s kind suggestion.  These four references regarding to in silico analysis have been included in reference 20-23 in the revised manuscript.

Please see the attachment, we revised the manuscript.

best regards

Samuchaya Ngamsuk 

Reviewer 3 Report

The authors prepared an hydrolysate of GAC Seed, they isolated different portions by HILIC and RP-HPLC and measured the ACE-inhibitory Activity.

From the most potent extract, they isolate and characterized a pool of low MW peptides, and finally they once again evaluate their ACE-inhibitory activity.

After this procedure they found ALVY peptide as the most active one, and they study the inhibitory mechanism by Lineweaver-Burk plot and they propose posible complexes after docking simulations.

They carried out a systematic study, they have experience in the field, reflected by several publications, however I consider that there are some topics to correct in order to improve the quality of the manuscript to be considered for publication in this journal.

I’ve separated my comments in two parts, in the first are the most relevant comments, at the end some minor mistakes.

Mayor issues:

  1. Unify the format of the references. References 2, 3 4 and 5 all have different formats.

       Reference 1 seems to be wrong, because is not related with the text. It should be related to the use of Gas in industry, and it refers to date palm seeds.

  Reference 3 does to correspond to medicinal use ose natural products extracted from Gac Seed.

Reference 4 is about the oil present in Gas Seeds, and not about their peptides

Reference 13 - Include the year.

2. In the first paragraph on page 2, lines 59 to 66 the authors describe the importance of ACE enzyme. In this text references 11 and 12 are included. However this references are about antihypertensive properties of peptides hydrolysates. Please use the primary source of information instead. Include references related to ACE instead.

3. In this paragraph the author stated:

“… Natural bioactive peptides from protein-rich foods provided an interesting explanation for ACE-inhibition and have been explored extensively as replacement of chemical drugs such as captopril, enalapril, and lisinopril.”

However there not references reflecting that the topic has been extensively explored.

4. On page 3, lines 102 and 103. “Other chemicals used in this experiment were of analytical grade” Which ones?

5. In this experimental  section, there is no description of the employed equipment.

6. On sections 2.2 and 2.3 the extraction and purification of some extracts is described. Which is the amount of recovered, or purified material at the end of those steps.

7. On page 6, Section 2.11. Water molecules were removed from the original X-ray structure, but chloride ions were kept. This is not usual, why these anions were retained?

8. On page 6, Section 2.12. What is HHL? This is the enzyme substrate?

9. One comment related with the IC50 determinations.
the IC50 of the extract is (70 +- 3.9) ug/mL. As you inform the results, I think that only one significative figure should be used in the error, and hence your measure should end in the same place, so (70 +- 4) ug/mL should be the correct way of representing this result.

10. On figures 2 and 3B there are some letter over the bars of the graphics. What is the meaning of these letters??

11. On page 9, the authors describe the methodology employed for the assignation of the sequence of the purified peptides. One of the compounds is ALVY, how do you confirm this peptide instead of, AVLY. Is easy to confirm the connectivity of the AA?

12. On page 10, authors said that the peptides were identified after a comparison with the signal of synthetic peptides. Please include their spectra. At least for the reviewing process.

13. On page 12 are presented the graphics for the determination of the IC50. values. The scale of all the graphics is the same, the reported values could be obtained from the graphics. In the case of LSTSTDVR, the 50% of inhibition is outside the presented range. This extrapolation is not correct.

14. One question about the activity essays. The different ratees of ACN in the different fractions.was considered in the determination of the inhibitory values of the compounds.

15. I don’t understand why after the identification and separation of Fraction 8 by HPLC (figure 3A), in a second HPLC separation a lot of peptides were detected.

16. The figure 6 is not clear, better figures should be included.
About the docking calculations, did you validate them docking an x-ray structure, or a well known inhibitor.

17. In Biopep there is a similar peptide, LVY, with a very good inhibition rate constant. You could also include it in the docking studies to enrich the discussion.

18. Related to the docking studies:

In the abstract author said:

“The molecular docking studies revealed that the ACE inhibitory activities of ALVY is due to interaction with the S1 (Ala354, Tyr523) and the S2 (His353, His513) pockets of ACE. Based on the Lineweaver-Burk plot, ALVY inhibited ACE competitively, which is consistent with the result observed in the molecular docking study.”

The real results are the experimental ones, docking studies propose a model that could help to explain your results. The results of docking could be consistent with the Lineweaver-Burk plot, and not in the oposite way.

A similar comment is included in the main text. I suggest to include the Lineweaver-Burk plot before the docking analysis.

Minor issues:

Page 3 - line 107 is spray dried instead of tray-dried?

Page 3 - lines 116 and 117- I think that “centrifuged at room temperature, 15,000xg for 10 min.” is duplicated.

Page 4 - Line 170 - What is ddH2O?

In parts of the document you acetonitrile, in other ACN, unify.

On page 5, line 218, there is webpage cited, is in the correct format?

On Page 5, line 226, What is the meaning of NNN?

Author Response

Author’s Reply to the Review Report (Reviewer 3)ส่วนบนของฟอร์ม

Major issues:

Comment 1: Unify the format of the references. (1) References 2, 3 4 and 5 all have different formats. (2) Reference 1 seems to be wrong, because is not related with the text. It should be related to the use of Gas in industry, and it refers to date palm seeds. (3) Reference 3 does to correspond to medicinal use ose natural products extracted from Gac Seed. (4) Reference 4 is about the oil present in Gas Seeds, and not about their peptides. (5) Reference 13 - Include the year.

Response: Thanks for Reviewer’s correction. They have been revised accordingly as follows:

  • The format of all references has been unified.
  • Reference 1 was replaced by “ Abdulqader, A.; Ali, F.; Ismail, A.; Mohd Esa, N. Antioxidant compounds and capacities of gac (Momordica cochinchinensis Spreng) fruits. Asian Pac. J. Trop. Biomed. 2019, 9(4), 158-167”.
  • Yes, reference 3 indicated the health benefits may be caused by natural products instead of active peptides. To avoid the misunderstanding, we removed the sentence “A possible reason for these healing properties would be due to the high content of proteins in the seeds [4]. Proteins and bioactive peptides play important roles in the metabolic functions of living organisms and, consequently, in human health [5].”, and reference 4 & 5. We also add a description “To explore more health benefits of gac seeds, this study was aimed to investigate the antihypertensive peptides from enzymatic hydrolysate of gac seeds.”
  • It was revised as above.
  • The year (1992) has been included in reference 14 (The reference 13 has been change to the reference 14).

Comment 2: In the first paragraph on page 2, lines 59 to 66 the authors describe the importance of ACE enzyme. In this text references 11 and 12 are included. However this references are about antihypertensive properties of peptides hydrolysates. Please use the primary source of information instead. Include references related to ACE instead.

Response: Thanks for Reviewer’s kind comments. The lines 59 to 66 have been change to line 58 to 65 and the references 11 and 12 have been change to the references 9-11.They have been revised accordingly as follows:

Reference 9 was replaced by " Daien, V.; Duny, Y.; Ribstein, J.; Du, C.G.; Mimran, A.; Villain, M.; Daures,

J.P.; Fesler, P. Treatment of hypertension with renin-angiotensin system inhibitors and renal dysfunction:

A systematic review and meta-analysis. Am. J. Hypertens. 2012, 25, 126-132“

Reference 10 was replaced by “Hollengerg, N.K. The renin-angiotensin system and cardiovascular disease. Blood

Press. 2000, 1, 5-8“

Reference 11 was replaced by “Erdos, E.G. Angiotensin I converting enzyme. Cir. Res. 1975, 36, 247-255“

Comment 3: In this paragraph the author stated:

“… Natural bioactive peptides from protein-rich foods provided an interesting explanation for ACE-inhibition and have been explored extensively as replacement of chemical drugs such as captopril, enalapril, and lisinopril.” However there not references reflecting that the topic has been extensively explored.

Response: Thanks for Reviewer’s kind comments. We add two references for this sentence as follows:

Reference 12:  Donkor, O.N.; Henriksson, A.; Singh, T.K.; Vasiljevic, T.; Shah, N.P. ACE-inhibitory activity of probiotic yoghurt. Int. Dairy J. 2007, 17, 1321-1331.

Reference 13: Beltrami, L.; Zingale, L.C.; Carugo, S.; Cicardi, M. Angiotensin-converting enzyme inhibitor-related angioedema: How to deal with it. Expert Opin. Drug Saf. 2006, 5, 643-649.

Comment 4: On page 3, lines 102 and 103. “Other chemicals used in this experiment were of analytical grade” Which ones?

Response: Thanks for Reviewer’s kind comments.  The lines have been change to the line 101 and 102. Some common chemicals used in this study were of analytical grade. For example: HCl, NaOH, hexene, boric acid, and so on.

Comment 5: In this experimental section, there is no description of the employed equipment.

Response: Thanks for Reviewer’s kind comment. The descriptions including the brand, model, maker, maker’s location and country, for the major instruments were already existed. The missing description for some minor equipment has been added.

Comment 6:  On sections 2.2 and 2.3 the extraction and purification of some extracts is described. Which is the amount of recovered, or purified material at the end of those steps.

Response: Thanks for Reviewer’s kind comment. The sections 2.2 Preparation and extraction of gac seed protein powder for purified protein in the extraction step. The amount of dried weight started at defatted gac seed power 60 mg for extracted protein and the yield of protein was 7 mg; the yield calculated according to the following equation:

%Yield = (Output/Input)*100% = (7 mg/ 60 mg)*100% = 11.66%  

For the sections 2.3, Gac seed protein hydrolysate for purified small peptides (molecular weight < 3 kDa) in the hydrolysate step. The amount of dried weight started at crude protein pellet 10 mg for hydrolysate and the yield of peptides (MW < 3 kDa) was 3.2 mg; the yield calculated according to the following equation:

%Yield = (Output/Input)*100% = (3.2 mg/ 10 mg)*100% = 32%

Comment 7: On page 6, Section 2.11. Water molecules were removed from the original X-ray structure, but chloride ions were kept. This is not usual, why these anions were retained?

Response: Thanks for Reviewer’s kind comment. Due to the reason that the zinc ion and chloride ion are two crucial cofactors of ACE, the chloride ion should be remained.

Comment 8:  On page 6, Section 2.12. What is HHL? This is the enzyme substrate?

Response: Thanks for Reviewer’s kind comment. The HHL is the substrate of the in vitro ACE inhibitory assay. The full name of HHL, hippuryl-L-Leucine, was included in Section 2.1. The reaction of this assay is shown below.

Hippuryl-Histdyl-Leucine has activities with ACE and water produced Hippuryc acid + Histidyl-Leucine

Comment 9: One comment related with the IC50 determinations. the IC50 of the extract is (70 +- 3.9) µg/mL. As you inform the results, I think that only one significative figure should be used in the error, and hence your measure should end in the same place, so (70 +- 4) ug/mL should be the correct way of representing this result.

Response: Thanks for Reviewer’s kind suggestion. It has been revised accordingly.

Comment 10:  On figures 2 and 3B there are some letter over the bars of the graphics. What is the meaning of these letters??

Response: Thanks for Reviewer’s kind comment. The letter over the bars of the graphics is a statistic annotation.  The different letters are significantly different at p < 0.05. The description was included in the original figure legends.

Comment 11:  On page 9, the authors describe the methodology employed for the assignation of the sequence of the purified peptides. One of the compounds is ALVY, how do you confirm this peptide instead of, AVLY. Is easy to confirm the connectivity of the AA?

Response: Thanks for Reviewer’s kind comment. Since the genome of Gac (Momoridica cochinchinensis Spreng.) has not been decoded completely, the peptide sequences were identified using LC-MS/MS coupled with de novo sequencing. Based on the peptide’s MS/MS spectrum (Figure 4), the automatic assignment of b and y ions was achieved using the software PEAK Studio 8.0. The sequences identified de novo were further confirmed using the synthetic peptides by comparing their retention time, MS and MS/MS spectra.

Comment 12: On page 10, authors said that the peptides were identified after a comparison with the signal of synthetic peptides. Please include their spectra. At least for the reviewing process.

Response: Thanks for Reviewer’s kind comment. The MS spectra of these four peptides are shown below.

ALVY

LLVY

LLAPHY

LSTSTDVR

Comment 13:  On page 12 are presented the graphics for the determination of the IC50. values. The scale of all the graphics is the same, the reported values could be obtained from the graphics. In the case of LSTSTDVR, the 50% of inhibition is outside the presented range. This extrapolation is not correct.

Response: Thanks for Reviewer’s kind comment. Indeed, the use of extrapolation for the IC50 calculation of LSTSTDVR was not suitable. We thus describe its IC50 as IC50 >> 166.67 µM (the maximum concentration in this assay; 166.67 µM).

Comment 14:  One question about the activity essays. The different ratees of ACN in the different fractions was considered in the determination of the inhibitory values of the compounds.

Response: Thanks for Reviewer’s kind comment. The fractions collected at different ACN percentage were accumulated for several repeated fractionation runs. All peptide fractions were freeze-dried and each fraction was prepared at 1 mg/mL of peptide concentration for in vitro ACE inhibitory assay. Based on the same peptide concentration, we can evaluate the ACE inhibitory activity of each fraction.

Comment 15: I don’t understand why after the identification and separation of Fraction 8 by HPLC (figure 3A), in a second HPLC separation a lot of peptides were detected.

Response: Thanks for Reviewer’s kind comment. The flow-through fraction roughly separated by a HILIC solid-phase extraction (SPE), instead of HPLC, was further separated by RP-HPLC. Due to the sample’s high complexity, the most active fraction (HILIC-FT/RP-F8) still contained mixture of peptide. Besides, LC-MS/MS is a very sensitive analytical tool which can also identify the peptides with very low abundance. Therefore, total 14 peptides were identified in this fraction.

Comment 16: The figure 6 is not clear, better figures should be included. About the docking calculations, did you validate them docking an x-ray structure, or a well known inhibitor.

Response: Thanks for Reviewer’s kind suggestion. It has been replaced using a higher quality figure. For the docking calculation, we did validate the docking at active site of PDB: 108a (the crystal structure of human ACE complex) using a commercial inhibitor lisinopril.

Comment 17: In Biopep there is a similar peptide, LVY, with a very good inhibition rate constant. You could also include it in the docking studies to enrich the discussion.

Response: Thanks for Reviewer’s kind suggestion. The potent ACE inhibitory peptide LVY (IC50= 5.84 µM) was not derived from any related food sources in the references included in this manuscript. It’s a little bit abrupt when discussed with it. The LVY has the same C-terminal sequence as our best inhibitor ALVY (IC50= 7.04 µM) and both peptides showed similar inhibitory activity. Hence, we prefer not to have docking comparison in this manuscript. But we think it is worthwhile studying the sequence-activity correlation based on the sequence of C-terminal LVY in our future study.

Comment 18:  Related to the docking studies:

In the abstract author said:

“The molecular docking studies revealed that the ACE inhibitory activities of ALVY is due to interaction with the S1 (Ala354, Tyr523) and the S2 (His353, His513) pockets of ACE. Based on the Lineweaver-Burk plot, ALVY inhibited ACE competitively, which is consistent with the result observed in the molecular docking study.” The real results are the experimental ones, docking studies propose a model that could help to explain your results. The results of docking could be consistent with the Lineweaver-Burk plot, and not in the oposite way.

Response: We agree Reviewer’s comment. The description was restated as “The inhibition kinetics study using Lineweaver-Burk plot indicated that ALVY is a competitive inhibitor. The inhibition mechanism of ALVY against ACE was further rationalized through the molecular docking simulation, which revealed that the ACE inhibitory activities of ALVY is due to interaction with the S1 (Ala354, Tyr523) and the S2 (His353, His513) pockets of ACE.”

Comment 19: A similar comment is included in the main text. I suggest to include the Lineweaver-Burk plot before the docking analysis.

Response: Thanks for Reviewer’s kind suggestion. In Section 3.6, we simulated the interaction between ACE binding site and the four small peptides using molecular docking. In Section 3.7, we only focused on the inhibition kinetics of ALVY against ACE. To keep the manuscript’s coherence, we prefer to keep the current arrangement.

Minor issues:

Comment 1: Page 3 - line 107 is spray dried instead of tray-dried?

Response: Thanks for Reviewer’s comment.  Yes, the gac seeds were tray-dried in this experiment.

Comment 2: Page 3 - lines 116 and 117- I think that “centrifuged at room temperature, 15,000xg for 10 min.” is duplicated.

Response: Thanks for Reviewer’s suggestion. It has been removed.

Comment 3: Page 4 - Line 170 - What is ddH2O?

Response: Thanks for Reviewer’s comment. The ddH2O should be double distilled water, but here it is regarded as deionized water generated using the PURELAB® water purification system from ELGA LabWater.

Comment 4: In parts of the document you acetonitrile, in other ACN, unify.

Response: Thanks for Reviewer’s suggestion. The full name (acetonitrile) was appeared in the Experimental section accompanied with its abbreviation (ACN). After that, all were unified as ACN.

Comment 5: On page 5, line 218, there is webpage cited, is in the correct format?

Response: it was revised as http://www.uwm.edu.pl/biochemia/index.php/pl/biopep.

Comment 6: On Page 5, line 226, What is the meaning of NNN?

Response:  The full name of NMM is 4-methylmorpholine. It was mentioned in Section 2. Materials and Methods.

Please see the attachment, some of the figure can not shown in here and I'm revised the manuscript but can not upload in here.

best regards

Samuchaya Ngamsuk

This manuscript is a resubmission of an earlier submission. The following is a list of the peer review reports and author responses from that submission.